# Effect of Transmission and Vaccination on Time to Dominance of Emerging Viral Strains: A Simulation-Based Study

**DOI:** 10.3390/microorganisms11040860

**Published:** 2023-03-28

**Authors:** Miguel Fudolig

**Affiliations:** School of Public Health, University of Nevada, Las Vegas, NV 89119, USA; miguel.fudolig@unlv.edu

**Keywords:** infectious disease modeling, emergent viruses, epidemic modeling, stochastic models, simulation-based experiments, vaccination, cross-immunity

## Abstract

We studied the effect of transmissibility and vaccination on the time required for an emerging strain of an existing virus to dominate in the infected population using a simulation-based experiment. The emergent strain is assumed to be completely resistant to the available vaccine. A stochastic version of a modified SIR model for emerging viral strains was developed to simulate surveillance data for infections. The proportion of emergent viral strain infections among the infected was modeled using a logistic curve and the time to dominance (TTD) was recorded for each simulation. A factorial experiment was implemented to compare the TTD values for different transmissibility coefficients, vaccination rates, and initial vaccination coverage. We discovered a non-linear relationship between TTD and the relative transmissibility of the emergent strain for populations with low vaccination coverage. Furthermore, higher vaccination coverage and high vaccination rates in the population yielded significantly lower TTD values. Vaccinating susceptible individuals against the current strain increases the susceptible pool of the emergent virus, which leads to the emergent strain spreading faster and requiring less time to dominate the infected population.

## 1. Introduction

Emerging viral strains in a population are not uncommon events [1,2,3], a was observed in influenza and SARS-associated coronavirus [4,5], Chikungunya [6], and Mayaro [7]. The phenomenon remained a mystery to the general public until the COVID-19 pandemic, where multiple variants of the COVID-19 virus (SARS-CoV-2) emerged in the population throughout the pandemic. Some examples of these variants are the Delta B.1.617.2, Omicron BA.1, BA.1.1, BA.2, BA.3, BA.4 and BA.5 [8]. Recent studies on the mathematical modeling of the spread of COVID-19 involves accounting for multiple strains in the population. Khyar and Allali [9] investigated a multi-strain SEIR (Susceptible-Exposed-Infected-Recovered) model and applied it to the COVID-19 pandemic. Fudolig and Howard [10] devised a modified SIR (Susceptible-Infected-Recovered) model that describes the dynamics of an emerging virus that resists available immunity to the population using a modified SIR model with vaccination. Otunuga [11] presented another approach to analyze the spread of emergent viral mutations in a population vaccinated against older strains, which was applied to analyze COVID infection data from Brazil and the United Kingdom. Massard et al. [12] analyzed COVID data from France by accounting for asymptomatic individuals in the multi-strain model. Li et al. [13] investigated the stability and equilibrium points of a two-strain epidemic model with single-strain vaccination in a complex network. Stochastic epidemic models have also been considered in studying the effect of vaccination in a population infected by multiple strains. For example, Chang and Liu [14] introduced standard Brownian motions in the surveillance data in a modified SIR framework to investigate the effect of vaccination in disease transmission. Mu and Zhang [15] studied near-optimal control for a stochastic multi-strain epidemic model with age structure and Markovian switching. These studies focused on developing optimal control methods to impede the spread of the different strains of the virus in the population, modeling the incidence data of the total infected population, or investigating parameter values that lead to the dominance of a specific strain in the population. One of the important aspects of a pandemic, that is often overlooked by researchers, is the time required for an emerging virus to account for more than half the infections in the infected population, which we refer to as time to dominance (TTD).

In the COVID-19 pandemic, emerging variants of SARS-CoV-2 were classified into three groups defined by the World Health Organization (WHO): Variants under Monitoring (VUM), Variants of Interest (VOI), and Variants of Concern (VOC) [16]. The working definition of VUMs were listed by the WHO as variants that “are suspected to affect virus characteristics with some indication that it may pose a future risk, but evidence of phenotypic or epidemiological impact is currently unclear, requiring enhanced monitoring and repeat assessment pending new evidence [16]”.

VUMs can easily be added/removed depending on how fast the variants evolve. When there is an increase in transmission or prevalence in multiple countries, variants are classified as a VOI. According to WHO, an emerging SARS-CoV-2 variant is classified as a VOI when it has “genetic changes that are predicted or known to affect virus characteristics such as transmissibility, disease severity, immune escape, diagnostic or therapeutic escape; AND identified to cause significant community transmission or multiple COVID-19 clusters, in multiple countries with increasing relative prevalence alongside increasing number of cases over time, or other apparent epidemiological impacts to suggest an emerging risk to global public health [16]”.

On the other hand, a variant is classified as a VOC if it satisfies the conditions for VOIs and an “increase in transmissibility or detrimental change in COVID-19 epidemiology; OR increase in virulence or change in clinical disease presentation; OR decrease in effectiveness of public health and social measures or available diagnostics, vaccines, therapeutics [16]”.

As of 26 February, 2023, the currently circulating VOCs were the Omicron variants and sublineages, which were classified as VOCs within a month of documenting the earliest samples. Previously circulating VOCs include the Delta B.1.617.2 variants and the Alpha B.1.1.7 variants, which were classified as VOCs within three and six months, respectively [16]. These VOCs account for the majority of cases in the infected population, as evidenced by genomic surveillance in the United Kingdom (https://coronavirus.data.gov.uk/details/cases (accessed on 23 March 2023)) and United States (https://covid.cdc.gov/covid-data-tracker/#variant-proportions (accessed on 23 March 2023)). The definition emphasizes that VOC classifications involve a decrease in effectiveness in control measures, including vaccines, in addition to an increase in transmissibility. Dominance of a new VOC can lead to changes in control strategies; hence. it is highly important to study what factors can affect the time required for an immunization-resistant emergent strain that satisfies the VOC requirements.

Although there has been extensive work on vaccination strategies to contain the spread of viruses [17,18,19,20,21,22,23,24,25], there has not been any research on how vaccination affects the emergence of immunity-resistant strains or variants of an existing virus in a population. This innovative study is the first of its kind in studying time to dominance (TTD) of an emergent strain in a population usinga compartmental model framework. We aimed to test whether the transmissibility of an emergent strain. has an effect on the time required for the emergent strain to dominate the infected population. The spread of the emergent strain was modeled using a stochastic, modified multi-strain SIR epidemic model, which is also a notable contribution in the stochastic modeling of multi-strain epidemics. The emergent strain was assumed to be resistant to the immunity through vaccine and previous infection of the existing dominant strain. Experiments performed on the simulations suggested that larger vaccination coverage and higher vaccination rates before, and during, emergence resulted in faster domination of the emergent strain. In addition, we determined whether there is a non-linear relationship between the relative transmission coefficient of the emergent strain and TTD for different values of vaccination rate and coverage. We focused on the implications of transmissibility and vaccination on the spread of an emergent strain on a population level, which is a limitation of compartmental models in epidemic modeling. The immunity provided by vaccination against the existing viral strain was assumed to be complete and not to wane in the duration of the simulations. Vaccination effects on an individual level, i.e., within-host effects, is beyond the scope of this study. In addition, the model follows a specific discretization scheme of a continuous time Markov chain presented by Breto et al. [26]. Other discretization and numerical schemes, such as epidemic branching processes [27] and Ito stochastic differential equations [28,29] that use Wiener processes in introducing stochasticity to the compartment transitions, are not considered in this study.

The flow of the paper is as follows: Section 2 provides the details of the stochastic SIR model, measurement of TTD, and implementation of the simulation-based experiment to test transmission and vaccination effects on TTD. Subsequently, Section 3 includes the key findings of the simulation-based experiments. Finally, Section 4 discusses the results presented in Section 3 and presents the appropriate conclusions based on the data.

## 2. Materials and Methods

The Methods section of this paper is divided into three parts: simulating the stochastic multi-strain SIR model with cross-immunity, based on the model presented by Fudolig and Howard [10]. The time to dominance (TTD) is calculated for each simulation the factorial experiment conducted. All simulations and analysis were performed using R statistical software [30].

### 2.1. Stochastic Multi-Strain SIR Model with Cross-Immunity

#### 2.1.1. Modified Multi-Strain SIR Model

The compartmental model used in the study is based on the modified multi-strain SIR model developed by Fudolig and Howard [10]. A diagram of the compartments and possible transitions is shown in Figure 1. Prior to the emergence of the new strain, the population is assumed to be infected by an existing viral strain and it is assumed that there are available vaccines that build complete immunity to the existing strain. The emergent strain is assumed to be completely resistant to the vaccine and capable of reinfecting individuals who recovered from the existing strain. We also assume that there is no direct super-infection or co-infection by the emergent strain, i.e., an individual cannot be infected by the emergent strain while infected by the existing strain. This behavior was found to be rare in SARS-CoV-2 and Influenza viral infections [31,32,33,34].

The coefficients β and $\beta^{\prime }$ are the respective transmission coefficients of the existing and emerging viral strains. The respective recovery coefficients for the existing and emerging viral strains are given by γ and $\gamma^{\prime }$. The rate ν is the vaccination rate upon the emergence of the new strain. The *S* compartment represents the susceptible population, *V* represents the vaccinated population, $R_{1}$ represents the individuals who recovered from the existing strain who are now susceptible to the emergent strain, $I_{1}$ represents the infectious individuals who are infected by the existing strain, $I_{2}$ represents the infectious individuals who are infected by the emerging strain, and $R_{2}$ represents the individuals who recovered from the emergent strain. It is assumed that recovery from the emergent strain provides complete immunity to both strains. The lowercase Latin letters denote the proportion of the population in each corresponding compartment. The population is assumed to be closed, and, hence, without the loss of generality, the birth and death rates were set to zero. The transition rates are governed by the law of mass action and dictated by the following differential equations: (1)$$\begin{array}{cc} \frac{ds}{dt} & =-\left[\nu s+\beta si_{1}+\beta^{\prime }si_{2}\right], \end{array}$$(2)$$\begin{array}{cc} \frac{dv}{dt} & =\nu s-\beta^{\prime }vi_{2}, \end{array}$$(3)$$\begin{array}{cc} \frac{di_{1}}{dt} & =\beta si_{1}-\gamma i_{1}, \end{array}$$(4)$$\begin{array}{cc} \frac{dr_{1}}{dt} & =\gamma i_{1}-\beta^{\prime }r_{1}i_{2}, \end{array}$$(5)$$\begin{array}{cc} \frac{di_{2}}{dt} & =\beta^{\prime }\left(s+v+r_{1}\right)i_{2}-\gamma r_{2}, \end{array}$$(6)$$\begin{array}{cc} \frac{dr_{2}}{dt} & =\gamma r_{2}. \end{array}$$
The total number of transitions from any compartment *A* to another compartment *B* at any time *t* is described by a counting process $N_{AB}\left(t\right)$ such that,
(7)$$\begin{array}{cc} s(t) & =s\left(0\right)-\frac{N_{SI_{1}}+N_{SI_{2}}+N_{SV}}{N}, \end{array}$$
(8)$$\begin{array}{cc} v(t) & =v\left(0\right)+\frac{N_{SV}-N_{VI_{2}}}{N}, \end{array}$$
(9)$$\begin{array}{cc} i_{1}\left(t\right) & =i_{1}\left(0\right)+\frac{N_{SI_{1}}-N_{I_{1}R_{1}}}{N}, \end{array}$$
(10)$$\begin{array}{cc} r_{1}\left(t\right) & =r_{1}\left(0\right)+\frac{N_{I_{1}R_{1}}-N_{R_{1}I_{2}}}{N}, \end{array}$$
(11)$$\begin{array}{cc} i_{2}\left(t\right) & =i_{2}\left(0\right)+\frac{N_{SI_{2}}+N_{VI_{2}}+N_{R_{1}I_{2}}-N_{I_{2}R_{2}}}{N}, \end{array}$$
(12)$$\begin{array}{cc} r_{2}\left(t\right) & =r_{2}\left(0\right)+\frac{N_{I_{2}R_{2}}}{N}, \end{array}$$
where *N* is the total population number.

#### 2.1.2. Continuous-Time Markov Chain Multi-Strain SIR Model

We introduced stochasticity to the multi-strain SIR model in Section 2.1.1 by creating a continuous-time Markov chain for the counting process $N_{AB}$. In particular, we assigned a probability distribution to $\Delta N_{AB}\left(t+\delta \right)=N_{AB}\left(t+\delta \right)-N_{AB}\left(t\right)$, which counted the number of transitions between compartments A and B between times *t* and t+δ, where δ is a time increment. We followed the framework designed by Breto et al. [26] in developing the Markov chains for the counting processes, which can be expressed as
(13)$$\begin{array}{cc} P\left[N_{AB}\left(t+\delta \right)=N_{AB}\left(t+\delta \right)+1\right] & =\mu (t)A(t)\delta +o(\delta ) \end{array}$$
where μ(t) is the time-dependent coefficient based on the transition equations in Equations (1)–(6). We apply these to the compartments in the model starting with the susceptible compartment. Within a time interval (t,t+δ), there are four possible transitions that a susceptible individual can undergo: $S\to I_{1}$; $S\to I_{2}$; S→V; and S→S, where the last transition corresponds to the individual remaining susceptible after the interval. We can simulate the number of transitions $\left(\Delta N_{SI_{1}},\Delta N_{SI_{2}},\Delta N_{SV}\right)$ by implementing a numerical scheme, based on Euler approximations [26]. There is a strong argument to use Poisson distributions in modeling the counting process, but the unbounded nature of Poisson random variables could lead to negative population values after the transitions [35,36]. Cai and Xu [35] and Breto et al. [26] presented a way to utilize multinomial distributions in simulating continuous time Markov chains with more than two transition channels, which, inherently, sets the population of a specific compartment as the upper limit of the number of transitions. We, then, applied the multinomial scheme to the presented stochastic SIR model with the following probabilities [37]:(14)$$\begin{array}{cc} P_{SI_{1}} & =\left[1-\exp\left[-\left(\beta i_{1}+\beta^{\prime }i_{2}+\nu \right)\delta \right]\right]\left[\frac{\beta i_{1}}{\beta i_{1}+\beta^{\prime }i_{2}+\nu }\right], \end{array}$$(15)$$\begin{array}{cc} P_{SI_{2}} & =\left[1-\exp\left[-\left(\beta i_{1}+\beta^{\prime }i_{2}+\nu \right)\delta \right]\right]\left[\frac{\beta^{\prime }i_{2}}{\beta i_{1}+\beta^{\prime }i_{2}+\nu }\right], \end{array}$$(16)$$\begin{array}{cc} P_{SV} & =\left[1-\exp\left[-\left(\beta i_{1}+\beta^{\prime }i_{2}+\nu \right)\delta \right]\right]\left[\frac{\nu }{\beta i_{1}+\beta^{\prime }i_{2}+\nu }\right]. \end{array}$$

In the absence of mortality, all the other compartments only have two possible transitions within a time interval (t,t+δ). use a binomial implementation scheme, based on the work of Tian and Burrage [36], to simulate the continuous time Markov chains. The remaining probability distributions of the transitions within a time interval (t,t+δ) can be expressed as:(17)$$\begin{array}{cc} \Delta N_{I_{1}R_{1}}\left(t\right) & ∼Bin(Ni_{1}\left(t\right),1-\exp\left(-\gamma \delta \right)), \end{array}$$(18)$$\begin{array}{cc} \Delta N_{R_{1}I_{2}}\left(t\right) & ∼Bin(Nr_{1}\left(t\right),1-\exp\left(-\beta^{\prime }i_{2}\delta \right)), \end{array}$$(19)$$\begin{array}{cc} \Delta N_{VI_{2}}\left(t\right) & ∼Bin(Nv_{1}\left(t\right),1-\exp\left(-\beta^{\prime }i_{2}\delta \right)), \end{array}$$(20)$$\begin{array}{cc} \Delta N_{I_{2}R_{2}}\left(t\right) & ∼Bin(Ni_{2}\left(t\right),1-\exp\left(-\gamma^{\prime }\delta \right)). \end{array}$$
Note that the binomial, multinomial, and Poisson schemes agree when δ→0 [26,37], and, hence, the trends observed in the simulated infected surveillance data would be the same regardless of the assumed distribution of the transition. We discretized the time evolution of the Markov chain by setting δ=1 and simulated weekly surveillance data using Euler’s method. The values of the coefficients were adjusted to fit the temporal resolution (week) of the model. After the surveillance data was simulated, we calculated the total proportion of infected, *i*, by adding the proportion of existing and emergent strain proportions. The proportion of the emergent strain within the infected population, $I_{2}$, could then be calculated using the equation
(21)$$\begin{array}{c} I_{2}=\frac{i_{2}}{i}𝟙\left(i>0\right), \end{array}$$
where 𝟙(i>0) is the indicator function that is equal to unity when i>0 and zero otherwise. The emergent strain is said to dominate the infections in the population when $I_{2}>0.5$, which is a condition we use in calculating time to dominance.

### 2.2. Time to Dominance (TTD)

We were interested in calculating the time required for the emerging strain to dominate in the population, which we refer to as time to dominance (TTD). A representative plot of strain infection proportion in the infected population is shown in Figure 2. The proportion of the emergent strain within the infected population could be observed to follow a logistic growth curve, a behavior that was also observed in genome-sequenced data of emergent SARS-CoV-2 strains in regions of the United Kingdom [38].

First, we checked whether $I_{2}$ exceeded 0.5 within the time frame of the simulation. If it did, we calculated the TTD by fitting a logistic growth curve on $I_{2}\left(t\right)$, such that
(22)$$\begin{array}{c} I_{2}\left(t\right)=\frac{1}{1+\exp\left(-\frac{t-t_{1/2}}{\sigma }\right)} \end{array}$$
where σ is the scale parameter of the logistic curve. The parameter $t_{1/2}$ is the time value that satisfies $I_{2}\left(t=t_{1/2}\right)=0.5$. Since logistic growth curves are monotonically non-decreasing functions, we assumed that the estimated TTD was equal to $t_{1/2}$. However, there were some cases where $I_{2}\left(t\right)$ was still increasing in time at the end of the simulation, which suggested that there was no change in curvature in the $I_{2}\left(t\right)$ curve. Without a change in curvature, the logistic growth curve would not be an appropriate function to fit $I_{2}\left(t\right)$. If $I_{2}\left(t\right)$ exceeded 0.5, and no change in curvature was detected, we fitted the simulated $I_{2}\left(t\right)$ data to an exponential growth curve. We also applied the exponential growth curve fit in scenarios when $I_{2}\left(t\right)$ did not exceed 0.5 by the end of the simulation, but was monotonically increasing. We performed linear regression after applying a log-transformation on $I_{2}\left(t\right)$. The TTD was then estimated using the following equation:(23)$$\begin{array}{c} \mathrm{TTD}=\frac{\ln(0.5)-\ln(b)}{k} \end{array}$$
where *k* is the slope of the linear regression and *b* is the estimated value of the intercept in the linear regression analysis.

### 2.3. Simulations and Factorial Experiment

We used a 3×4×3×4 factorial design to analyze the effect of transmissibility and vaccination during the emergence of a new viral strain in an infected population. The experimental units were the simulated runs of the stochastic multi-strain model in Section 2.1. The factors included in the experiment were the following: transmission coefficient of the existing strain β; emergent strain transmission ratio (ESTR), i.e., relative transmission coefficient of the emergent strain $\beta^{\prime }/\beta$; vaccination rate ν; and initial vaccine coverage in the population v(0). The levels of each factor used in the experiment are shown in Table 1. The parameter values were selected after screening for parameter values that led to a steady increase in emergent strain infections in the population during the simulation. Note that the values of the ESTRs investigated in the experiment were greater than 1, which implied that the emergent strain was more transmissible than the existing strain. The recovery coefficients ($\gamma ,\gamma^{\prime }$) were set to 0.7 for both viral strains, which assumed that the infectious period for infected individuals was 10 days ≈ 1.42 weeks. The values for the transmission coefficient of the existing strain, β, were chosen such that the emergent strain was expected to dominate the population on its own, regardless of the availability of vaccines [17]. The vaccination rate ν=0 corresponded to the absence of vaccination in the susceptible population, while ν=0.5 and ν=0.25 corresponded to the cases where vaccinated individuals developed full immunity from the existing strain after 2 and 4 weeks, respectively. The values of the initial vaccine coverage, which was the proportion of vaccinated individuals before the new strain’s emergence, were based on the national influenza and SARS-CoV-2 vaccine coverage in the United States in 2023 [39,40]. The value v(0)=0.25 was added to ensure equal spacing between the factors for formal statistical analysis.

The initial proportion of recovered individuals ($r_{1}\left(0\right)$) was set at 0.01, while the initial proportion of the infected individuals (*i*) was set at 0.01. Upon emergence, we set the proportion of the infected by the emerging strain among the infected, ($I_{2}\left(0\right)$) as 0.01. Each run was simulated until t=12 weeks was reached with a time step size of δ=1 week. The choice of time step and maximum time value was based on whether the emerging strain could dominate the infected population within three months. A total of 10 simulated runs were performed for each factor combination and the TTD was calculated for each run. The data was analyzed using a linear model approach to test for interactions and simple effects. Least squares means (LSMeans) estimates and contrasts were calculated using the emmeans package in R [41]. A significance level of 0.05 was used in testing hypotheses and constructing confidence intervals.

## 3. Results

The simulations from the factor combinations considered in the study behaved as expected, aside from a few exceptions, as can be seen in Figure 2. We discovered that a large percentage of the runs for the factor combination (β,$\beta^{\prime }/\beta$,ν,v(0)) = (1.4, 1.25, 0, 0) resulted in an increasing trend in $I_{2}$, but failed toexceed 0.5 after 12 weeks. Figure 3 shows one of the runs for the aforementioned factor combination, where the $I_{2}=0.31$ when the simulation ended. These scenarios led to TTD values greater than 12, implying that the emergent strain would eventually dominate the infected population if the simulation was extended. The estimated TTD values for these scenarios were calculated using an exponential growth curve fit outlined in Section 2.2.

Simulations for higher values of β and ESTR exhibited behavior where both strains died out in the population before the simulation ended. This extreme behavior was caused by the complete recovery of infected individuals after all previously susceptible and immunized individuals had been infected by the two strains, as shown in Figure 4. Only the $I_{2}$ values before the sudden drop were considered in calculating the TTD values using a logistic growth curve fit.

### 3.1. Interaction Plots

We used a Type III test of fixed effects to test the main effects and interactions for the four factors considered, which yielded a significant four-way interaction (p<0.0001). Hence, we needed to examine how the TTD values behaved at each simulation and parameter combination. The interaction plot in Figure 5 shows the least squares means estimates plotted across the ESTR values and sliced by vaccination rates. Each panel corresponds to a different combination of v(0) and β.

Figure 5 shows how higher vaccination rates (ν) yielded in our simulations. This behavior was observed across all levels of transmissibility of both existing and emergent strains, as well as vaccination coverage levels. We could also deduce from the plots that, meaning wider vaccination coverage before emergence led to lower TTD values with less variability. When the existing strain was more transmissible, the time to dominance was observed to substantially decrease for populations with lower vaccination coverage (v(0)=0,0.25). In general, higher values of the ESTR yielded lower TTD values, but the differences between the ESTR levels decreased for higher vaccination rates and initial vaccination coverage. This behavior was observed across all β values, which led to the hypothesis that the ESTR has a non-linear relationship with TTD for select vaccination rates and coverage. These observations were formally tested using simple effects and orthogonal polynomial contrasts. The Bonferroni correction was applied to account for multiplicity in pairwise comparisons when necessary. The exact estimates, *p*-values, standard errors, and confidence intervals from the analysis of this paper can be accessed here: https://github.com/MiguelFudolig/SIRwithUK (accessed on 23 March 2023).

### 3.2. Vaccination Rate ν

The effects of vaccination rate were significant in all parameter combinations (p<0.05), except for the parameter combination β=1.4, ESTR = 2, and v(0)=0.75, which yielded a moderately significant effect (F(2,1296)=2.402,p=0.09). Upon examining the pairwise comparisons of vaccination rate levels, the difference in TTD between the populations with ongoing vaccinations (ν=0.25 and ν=0.5) were mostly non-significant at the 0.05 level, after applying the Bonferroni correction when the initial vaccination coverage was set at 0.75. The estimated differences ranged from 0.07 (95% CI: −0.22, 0.37) to 0.52 (0.23, 0.82) weeks, which were substantially lower than the differences observed for initially unvaccinated populations, where the TTD differences were estimated to be from 0.66 (0.37, 0.96) to 1.49 (1.19, 1.78) weeks. This result implies that when the vaccination coverage is already high in the population before the emergence of the new strain. The vaccination rate would not have a significant effect on the TTD values.

The average TTD value of populations with ongoing vaccinations were determined to be significantly lower compared to the populations without ongoing vaccinations (ν=0), with estimated differences ranging from 0.12 (−0.09, 0.33) to 0.46 (0.25, 0.68) weeks when the initial vaccination coverage was set at v(0)=0.75. The estimated differences were considerably higher for the initially unvaccinated populations (v(0)=0), where the estimated differences ranged from 2.55 (2.34, 2.75) to 9.89 (9.68, 10.1) weeks. These differences suggest that the presence of active vaccination measures led to faster dominance by the emergent strain in the infected population.

### 3.3. Initial Vaccination Coverage v(0)

The effect of the initial vaccination coverage was determined to be highly significant in all parameter combinations (p<0.01). Upon examining the pairwise effects, we determined that the highest TTD values occurred when there was no initial vaccination coverage (v(0)=0) in the population (adjusted p<0.01). Holding other parameters constant, the TTD values were observed to decrease as the initial vaccination coverage increased. The average difference in TTD between v(0)=0 and v(0)=0.75 was calculated to be 3.43 (3.37, 3.48) weeks. Comparing consecutive levels showed that an increase of 0.25 in initial vaccination coverage could lead to a decrease in TTD, ranging from 1.01 (0.96, 1.06) to 1.30 (1.25, 1.35) weeks. These results imply that vaccination coverage is negatively associated with time to dominance, regardless of vaccination rate and emergent strain transmissibility.

### 3.4. Relative Transmissibility of Emergent Strain (ESTR)

The effect of ESTR on TTD was highly significant in all but two treatment combinations (p<0.01): (β,ν,v(0)=1.4,0.5,0.75) with F(3,1296)=2.076, p=0.102 and (β,ν,v(0)=2.1,0.25,0.75) with F(3,1296)=2.177, p=0.089. It was noteworthy that these treatment combinations involved high vaccination rates and coverage. These effects were examined further by testing the pairwise TTD differences of consecutive ESTR levels, i.e., ESTR = 1.25 vs. ESTR = 1.5, ESTR = 1.5 vs. ESTR = 1.75, etc., for all levels of vaccination rates and coverage. We determined that the TTD differences between consecutive ESTR levels were not significant (p>0.06) when the initial vaccination coverage was set at 0.75, but were significant (p<0.05) otherwise. This result implied a non-linear relationship between ESTR and TTD for populations with low vaccination coverage because of how the slopes changed as the ESTR was increased. We tested with orthogonal polynomial contrasts for quadratic and cubic relationships between ESTR and TTD for different vaccination coverages and rates. When there was no vaccination in the population, the relationship between the TTD and ESTR values were cubic in nature ($t_{1296}=-2.35$, p=0.02). However, the relationship became quadratic or linear when ν or v(0) were non-zero. When the initial vaccination coverage was 0.75, only the linear effect of the ESTR was significant for all ν values (p<0.01). Quadratic effects were only significant for the following vaccination coverage and rates: ν=0.25 and v(0)=0 (p<0.01); ν=0 and v(0)=0.25 (p=0.049); and ν=0 and v(0)=0.5 (p<0.001). These results are supported by the plots in Figure 5, where the curvature is more apparent for lower vaccination rates and coverage. A slight increase in relative transmissibility of the emergent strain would lead to a drastic decrease in TTD value if the population had low vaccination coverage. On the other hand, the TTD values were found to be unaffected by changes in the ESTR when the population had high vaccination coverage.

## 4. Discussion

The results suggest that the relative transmissibility of the emergent strain, vaccination rate, and initial vaccination coverage of the population did have an effect on the time required for the emergent strain to dominate the infected population regardless of the transmissibility of the original strain. The Type III tests of fixed effects yieldeda significant four-way interaction between the four experimental factors, which meant that necessitated examing the simple effects and pairwise comparisons. This finding was consistent with the available multi-strain epidemic models cited in Section 1, where the stability and existence of emergent strain-dominant equilibrium points were dependent on the values of transmission coefficients and vaccination rate. However, our experiment provided more insight on how these coefficients affected the rate at which an emergent strain among the infected population could be achieved in the stochastic framework. Based on the results of simple effects tests and pairwise comparisons, we discovered that there was strong evidence that higher vaccination rates and higher pre-emergence vaccination coverage in the population led to lower TTD values, provided that the available vaccine gave individuals complete immunity from the existing strain. We also discovered that the effect of the vaccination rate on the TTD was non-significant when the proportion of vaccinated individuals was already high when the new strain emerged. Highly transmissible emergent strains were found to dominate the infected population faster than weakly transmissible emergent strains. The relationship between the TTD and the relative transmissibility of the emergent strain was found to be non-linear for populations with low vaccination coverage and linear for populations with high vaccination coverage, as evidenced in Figure 5. We. therefore, conclude, based on the results, that highly transmissible vaccine-resistant emergent viral strains or variants would dominate the infections in the population faster if there was a large proportion of vaccinated individuals at the time of emergence. Moreover, continuing vaccination efforts for populations with low vaccination coverage after the emergence of a vaccine-resistant emergent viral strain was found to lower the TTD value for the emergent viral strain or variant. The novel discoveries from this simulation-based experiment seem counter-intuitive at first because previous studies and public health advisories used vaccination as a strategy to control the the incidence of viral infections [5,11,17,18,19,20,21,22,23,24,25,42,43,44,45,46]. The rapid dominance of the emergent strain within the infected should not be perceived as the failure of a vaccine, but rather, it is a by-product of the effectiveness of the vaccine. In the presented stochastic epidemic model, the available vaccine was designed to contain the spread of the existing strain, not the emergent strain. Figure 2, Figure 3 and Figure 4 show that the proportion of existing strain infections also decreased rapidly when the emergent strain infections increased, which was consistent with previous findings of emergent strain endemic equilibria in vaccinated populations [10,11]. The vaccine helps control the spread of the existing strain in the population in the long term [10,17], but the same vaccine formulation does nothing to protect the population for a vaccine-resistant emergent viral strain. A larger vaccinated population provides the newly emergent viral strain with a larger “effective” susceptible pool that is inaccessible to the existing viral strain due to the immunity provided by vaccination and recovery. The access to individuals immune to the existing strain enables the emergent viral strain to spread and dominate faster in the population. On the other hand, the proportion of individuals susceptible to the current strain decreases rapidly due to vaccination. As a result the emergent strain infections from the emergent strain increase, while the current strain’s infections decrease. This behavior leads to a low TTD value regardless of how transmissible the emergent strain is, which is consistent with what we observed in Figure 5.

These conclusions have huge implications in vaccination policies in highly susceptible populations. When there is a threat of a highly infectious vaccine-resistant strain, or a newly detected VUM, emerging in the population, pushing for larger vaccination coverage in the population would lead to faster spread of the new variant in population. This result could be applied to highly infectious diseases, such as SARS-CoV-2 and Influenza, which have been recorded to have highly drug- and vaccine-resistant mutations and variants in recent years. These implications present a conundrum for policymakers in considering vaccination as a control strategy. Should policymakers endorse vaccination to control the spread of the currently dominant strain but leave the population vulnerable to a possible outbreak of a vaccine-resistant strain? The results of the experiment suggest that if a highly transmissible vaccine-resistant strain emerges when there is low vaccine coverage in the population, then it would be beneficial to suspend vaccination in the susceptible population the spread of the emergent strain until an updated vaccine developed. However, there is no benefit in suspending or continuing vaccinations after the emergence if there was already a high vaccine coverage in the population prior to the emergence of the new strain.

In summary, we used a simulation-based experiment to determine the effect of the relative transmissibility of an emergent strain and vaccination status of the population on the dominance of an emerging viral strain. There was strong evidence of a decrease in TTD when vaccinated individuals were present and vaccinations were administered to the susceptible population. We also have strong evidence of a negative association between the relative transmissibility of the emergent strain to the TTD. The negative relationship was discovered to be cubic for populations with no vaccination measures in place and linear for populations with high vaccination coverage and rates. These results were based on the assumptions of the stochastic SIR model presented in the paper, which were as follows: (1) the vaccine provides individuals with complete and unwaning immunity to the existing strain, and (2) all transition processes, such as infection, recovery, and vaccination, are dictated by Markov processes. The results of the statistical analysis may not be robust for all stochastic SIRV models and discretization schemes of the continuous time Markov chain, which are promising areas to investigate in future studies.

Utilizing a simulation-based experiment was a novel and innovative approach that gave us more insight into how rapidly an emerging viral strain could dominate in a population. This approach also allowed us to study the effect of vaccination and cross-immunity for various levels of transmission for both emergent and existing strains without waiting for a viral strain to emerge. It is highly recommended for future researchers to use this method in studying the spread of can easily be adapted to other SIR-like epidemic models that include exposed (SEIR) and quarantined compartments, provided that the models are stochastic in nature. In addition, our method can be used to test optimal control strategies in hypothetical situations. We recommend incorporating experimental design concepts in future work involving simulation-based experiments in infectious disease modeling.

## Figures and Tables

**Figure 1 microorganisms-11-00860-f001:**
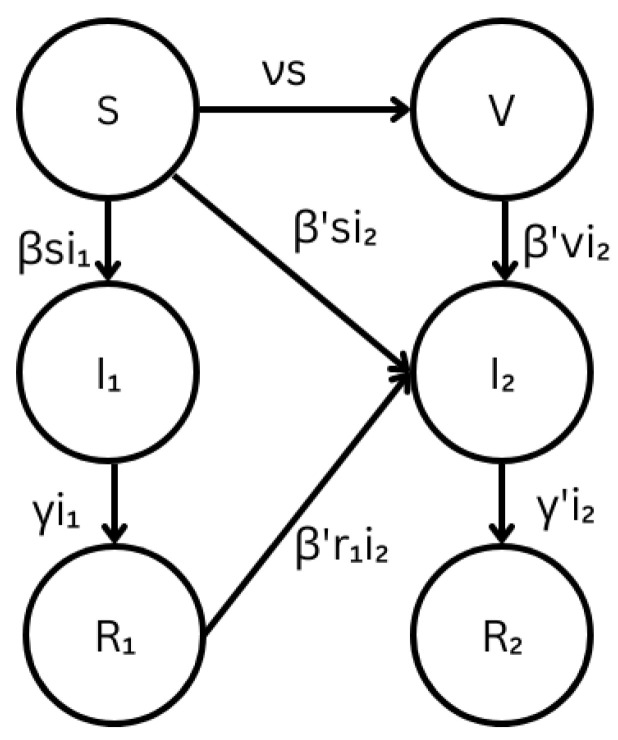
The schematic diagram of the multi-strain SIR model for emerging viral diseases with cross-immunity. The diagram includes the compartments and transition rates between compartments. Definitions of the transition coefficients are available in Table A1.

**Figure 2 microorganisms-11-00860-f002:**
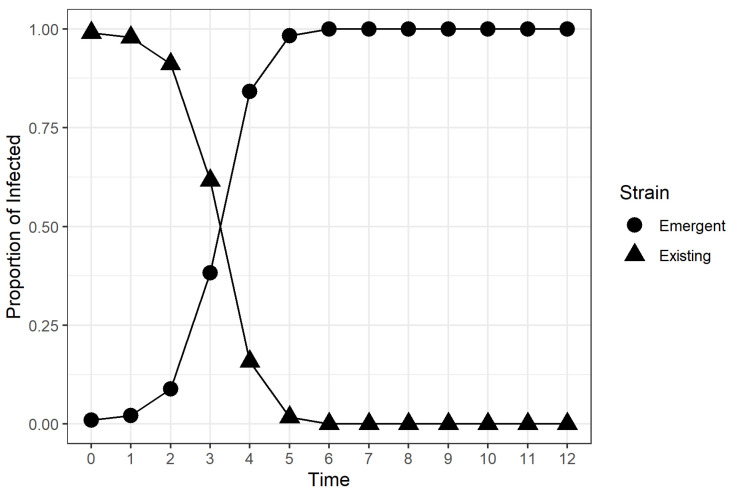
A representative simulation run for the factor combination (β, $\beta^{\prime }/\beta$, ν, v(0)) = (1.4, 1.5, 0.5, 0.25). The proportion of emergent strain infections follows a logistic growth curve.

**Figure 3 microorganisms-11-00860-f003:**
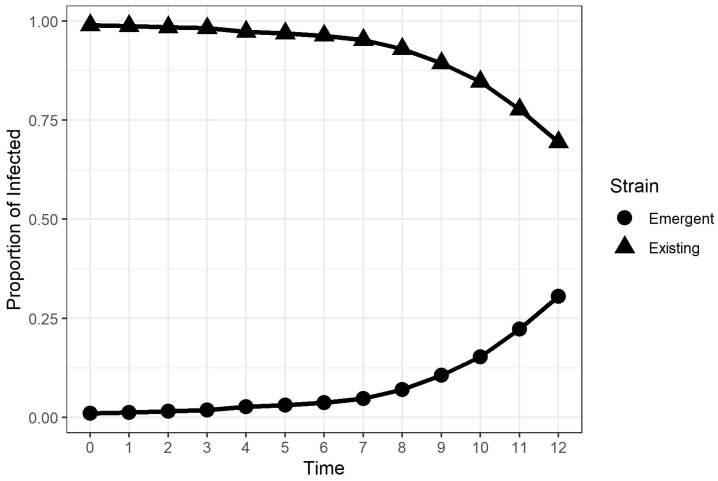
A sample run for the factor combination (β, $\beta^{\prime }/\beta$, ν, v(0)) = (1.4,1.25,0,0). The proportion of emergent strain infections among the infected did not go above 0.5, despite observing a steady increase in infections.

**Figure 4 microorganisms-11-00860-f004:**
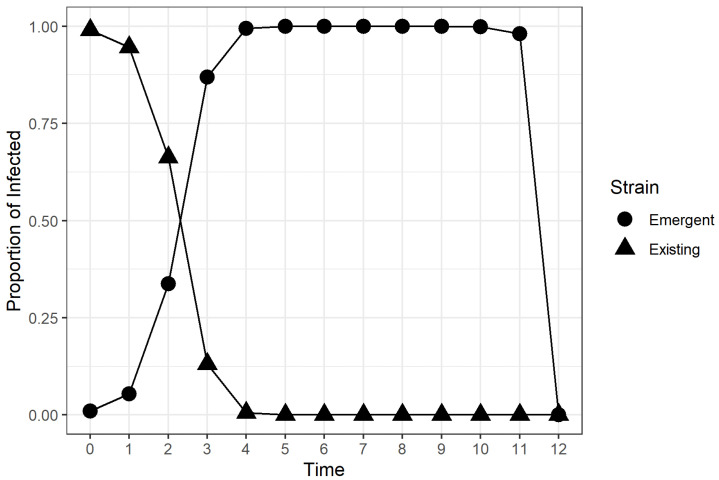
A sample run for the factor combination (β, $\beta^{\prime }/\beta$, ν, v(0)) = (2.1, 2, 0.5, 0.5). Both proportions dropped to zero at time t=12, due to complete recovery after complete infection of all infected individuals in the population.

**Figure 5 microorganisms-11-00860-f005:**
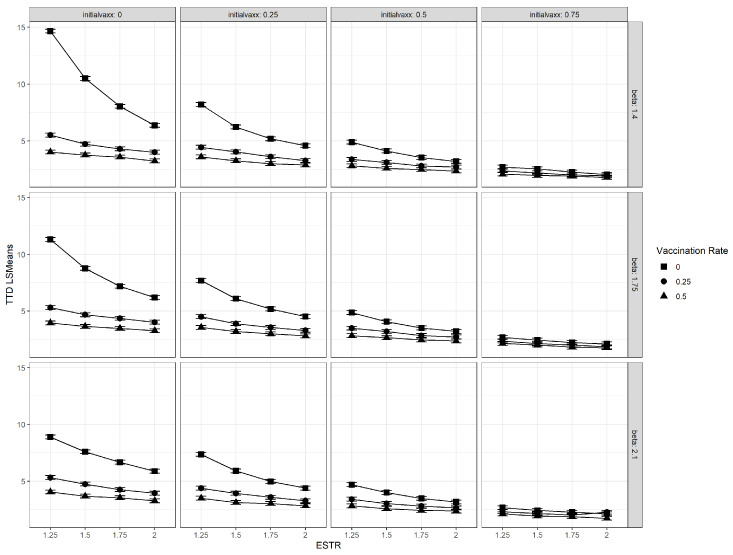
The interaction plots of the vaccination rate and emergent strain transmission ratio (ESTR) for different combinations of the initial vaccination coverage (initialvaxx), and transmission coefficient of the existing strain (beta).

**Table 1 microorganisms-11-00860-t001:** The levels of each factor in the simulation-based experiment.

Factor	Simulation Parameter	Values
Transmission coefficient of existing strain	β	1.4, 1.75, 2.1
Emergent strain transmission ratio (ESTR)	$\beta^{\prime }/\beta$	1.25, 1.5, 1.75, 2
Vaccination coefficient	ν	0, 0.25, 0.5
Initial vaccination coverage	v(0)	0, 0.25, 0.5, 0.75

## Data Availability

The code for simulation and data analysis is available on GitHub (https://github.com/MiguelFudolig/SIRwithUK) (accessed on 23 March 2023).

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
