# Peer review of "Effect of Transmission and Vaccination on Time to Dominance of Emerging Viral Strains: A Simulation-Based Study"

_microorganisms, 2023, doi:10.3390/microorganisms11040860_

Round 1

Reviewer 1 Report

I have nothing against this paper that, I anticipate, deserves publication if the issues below are adequately addressed.

Nonetheless, some factors remains obscure.

For example, one of the main findings of the paper is that, independent of the vaccination status  fo a population, an emergent strain will become seroprevalent very fast. This is very counterintuitive.

I suppose there could be be explanations (both at a mathematical and biological level) that should be discussed, and clearly emphasized as LIMITATIONS of the paper, for example:

1 the simulation presented in this paper, with its underlying stochastic model, has been not adequately designed. For example, there is scarce information about the distributions on which the SIR model is based. those distributions are neither clearly discussed nor motivated.

I take as example a paper where changing from a Poisson to Negative binomial did not change the results of the model which was robust enough to provide evidence in support of the hypothesis:

AA.VV. Reopening Italy’s schools in September 2020: a Bayesian estimation of the change in the growth rate of new SARS-CoV-2 cases.

Author Response

I am deeply grateful for your comments on this paper. Here are my responses (in bold face) to your comments and suggestions.

I have nothing against this paper that, I anticipate, deserves publication if the issues below are adequately addressed.

Nonetheless, some factors remains obscure.

For example, one of the main findings of the paper is that, independent of the vaccination status  fo a population, an emergent strain will become seroprevalent very fast. This is very counterintuitive. I suppose there could be be explanations (both at a mathematical and biological level) that should be discussed, and clearly emphasized as LIMITATIONS of the paper, for example:

1) the simulation presented in this paper, with its underlying stochastic model, has been not adequately designed. For example, there is scarce information about the distributions on which the SIR model is based. those distributions are neither clearly discussed nor motivated. I take as example a paper where changing from a Poisson to Negative binomial did not change the results of the model which was robust enough to provide evidence in support of the hypothesis:AA.VV. Reopening Italy’s schools in September 2020: a Bayesian estimation of the change in the growth rate of new SARS-CoV-2 cases. BMJ Open 2021;11:e051458. doi: 10.1136/bmjopen-2021-051458. I would like the authors discuss on this point, hopefully taking inspiration from the mathematical modeling of the above paper which was robust to distribution change.

  • Thank you for this helpful comment. I recognize that I did not include the justification on why I chose the multinomial and binomial distribution to simulate stochastic transitions in my model. The stochastic model uses continuous time Markov process that can be approximated by schemes using Poisson, binomial, and multinomial distributions. All these distributions are equivalent for low time steps. The main disadvantage of using the Poisson scheme in simulating transitions is that it could lead to negative proportions because of its unbounded nature. Hence, I used the binomial and multinomial schemes. I have included a brief explanation and sources referring to the schemes in Lines 149-158

2 I think that the results the authors have achieved can depend on the type of vaccination they have in mind. Maybe, different results could be achieved depending wether the vaccine provides a sterilizing immunity or not (like in the case for COVID). This point should be clearly discussed.

  • Thank you for this interesting comment. The intra-host mechanism of the vaccination applied in the population is beyond the scope of the study because I was interested in a macroscopic view of the spread of infections. The main assumption was that it provided complete and unwaning immunity from the current strain, regardless of how immunity was achieved. One factor that could be related to the type of vaccination is the vaccination rate, which is the rate at which vaccinated individuals develop immunity from the current existing strain. The effect of the vaccination rate is higher when there is low vaccine coverage and the emergent strain has relatively low transmissibility. Both the revised Results and Discussion sections discuss the effect of vaccination on the time to dominance of the emergent strain in further detail compared to the original manuscript.

Reviewer 2 Report

The manuscript describes a stochastic simulation study of emergence of novel viral strain in the presence of a vaccine for the existing strain. Overall, the paper is clear, the work appears to be scientifically sound, and the conclusions are supported by the results. I have a few suggestions for improving the manuscript:

1. I think it would be helpful to see some sample time courses from the simulations. Having a case with clear dominance of the new virus and one where the virus does not achieve dominance would help the reader understand the rest of the analysis.

2. My only real quibble with the methodology is the small number of parameter values used in the study. It's hard to judge whether we're getting the full picture of the possible behaviours. Does the author have some reasoning for choosing these particular values over these particular ranges?

3. The discussion is rather superficial --- how does this work tie into other work that is out there? The introduction mentions several other studies of emergence of a virus, and while they do not specifically measure TTD, are there other measures consistent with the results presented here? The author might also look into within-host viral models examining emergence of drug-resistant mutants, where TTD (or something similar) is calculated. Are there similar findings in this system?

4. When using et al., there is no period after et. (lines 31 and 32).

5. Line 129: I think B is supposed to be part of the subscript.

6. Equation 13 is a blank line.

7. There should not be a paragraph indent after an equation when continuing the sentence (lines 130, 138, 155, 163, 173)

8. Line  132: remove 'a'

9. Line 287: 'which' should be 'that'.

Author Response

Thank you for your helpful review. I have enclosed my responses to your comments and suggestions in bold face.

The manuscript describes a stochastic simulation study of emergence of novel viral strain in the presence of a vaccine for the existing strain. Overall, the paper is clear, the work appears to be scientifically sound, and the conclusions are supported by the results. I have a few suggestions for improving the manuscript:

  1. I think it would be helpful to see some sample time courses from the simulations. Having a case with clear dominance of the new virus and one where the virus does not achieve dominance would help the reader understand the rest of the analysis.

Thank you very much for this comment. I realized that I needed to add sample time courses to explain why I used logistic and exponential growth curves to measure the time to dominance. Figures 2, 3, and 4 of the manuscript show sample time courses of the simulations. Figure 2 shows a typical time course, Figure 3 shows when dominance does not occur within the 12 weeks but is expected to occur weeks later, and Figure 4 shows cases where there is a sudden drop in infected proportions because all susceptible and immunized individuals have been infected and recovered.

  1. My only real quibble with the methodology is the small number of parameter values used in the study. It's hard to judge whether we're getting the full picture of the possible behaviours. Does the author have some reasoning for choosing these particular values over these particular ranges?

This is a great question. One of the advantages of using a factorial design is we can test for different relationships and effects within the range dictated by the chosen levels of each factor. To address this comment, I added more values of vaccination coverage and vaccination rates to test the effect of vaccination on a wider scale. I chose the vaccination coverages based on the COVID-19 vaccination coverage (65-80%) and Influenza (~50%) in the United States, while the vaccination rates were chosen such that the non-zero vaccination rates correspond to immunity building times of 2 (0.5) and 4 (0.25) weeks from vaccination. The effects of the vaccination rates and coverages were thoroughly discussed in the revised Results and Discussion sections.

  1. The discussion is rather superficial --- how does this work tie into other work that is out there? The introduction mentions several other studies of emergence of a virus, and while they do not specifically measure TTD, are there other measures consistent with the results presented here? The author might also look into within-host viral models examining emergence of drug-resistant mutants, where TTD (or something similar) is calculated. Are there similar findings in this system?

Thank you for these comments. I have improved the discussion by emphasizing the implications of the results in implementing vaccination policies in populations and the risk of a rapid spread of a newly emerged immunization-resistant viral strain. There were other measures consistent with the results such as the existence of an emergent strain endemic equilibrium even in the presence of vaccination, and related works were cited in Lines 360-362.

The within-host characteristics of the virus infection is outside the scope of this paper, which I have now included as a limitation in the Introduction. To my knowledge, it is not surprising to find similar findings in recombination and mutation of viruses inside a host because these processes have been modeled using the law of mass action, which is the same law that dictates the transition between compartments in the presented stochastic SIR model. Although there are analogous concepts between the two scales of modeling, it would be misleading to relate the macroscopic and microscopic mechanisms of immunization due to the similarity in underlying simulation framework.

  1. When using et al., there is no period after et. (lines 31 and 32).
  2. Line 129: I think B is supposed to be part of the subscript.
  3. Equation 13 is a blank line.
  4. There should not be a paragraph indent after an equation when continuing the sentence (lines 130, 138, 155, 163, 173)
  5. Line 132: remove 'a'
  6. Line 287: 'which' should be 'that'.

Thank you very much for pointing out formatting and typographical errors in my manuscript. I have fixed them all in the revised manuscript.

Reviewer 3 Report

1) Author allows write en plural but there’s is only one author, please check it with a English specialist.   2) In introduction: eliminate lines 48 to 53; 56 to 63. Line 64 to 69 change redaction to avoid to cop textually the considerations.    3) Why is the author augment to asume not a direct superinfection by the emergent strain? And if the authors assume a superinfection event, how this can change the model and their results?   4) About the ata from table 1, where the author obtain it? Please add references or why they chose this values.    5) The result in lines 222-224 and 229-231, are ver interesting and crucial. Please provide describe a little better you results and how did you get to this important conclusion. Also in discussion, make a deeper argument of this result. 

Author Response

Thank you very much for your helpful review. Your feedback is greatly appreciated. I have enclosed my responses to your review in bold face.

1) Author allows write en plural but there’s is only one author, please check it with a English specialist.  Thank you very much for your comment on my writing style. I have consulted an English specialist as well as known technical writing sites, and the use of We is permitted even in single author papers. I have edited parts of the manuscript that used “our research” to “the research”.

 2) In introduction: eliminate lines 48 to 53; 56 to 63. Line 64 to 69 change redaction to avoid to cop textually the considerations. Thank you very much for your comment. I have moved the quotations in-line to consider the format of the text.

    3) Why is the author augment to asume not a direct superinfection by the emergent strain? And if the authors assume a superinfection event, how this can change the model and their results?

These are really good questions and I appreciate you asking them. Super-infection between variants of common viral infections such as influenza and SARS-CoV-2 is quite rare. I have added references that investigated superinfection and coinfection between variants in the manuscript (line 118-121). If we assumed that there is a superinfection event, I would say that the time to dominance would be lower because it gives the emergent strain another source of individuals susceptible to infection.

  4) About the ata from table 1, where the author obtain it? Please add references or why they chose this values.  

Thank you very much for your comments. The transmissibility coefficients were chosen based on values that would lead to an emergent strain endemic equilibrium, the vaccination rates were based on the time it takes to build immunity from common viral vaccines (Influenza and SARS-CoV-2), and vaccination coverage is based on the coverage of Influenza and SARS-CoV-2 in the United States. I have added references and justifications for these values in Section 2.3.

 5) The result in lines 222-224 and 229-231, are ver interesting and crucial. Please provide describe a little better you results and how did you get to this important conclusion. Also in discussion, make a deeper argument of this result.

Thank you very much for these comments. The conclusions from these lines were obtained from formal statistical analysis presented in the Results section. I have listed p-values and confidence intervals of relevant statistical tests in the revised Results sections, which can help point out the effects of transmission and vaccination on the time to dominance. I have added more details and discussed the implications of the results in further detail in the revised Discussion sections.

Round 2

Reviewer 1 Report

I noticed that no one of my previous annotations (i.e., distributions, limitations, references) have been taken into serious consideration and the paper has not changed in response to them. I cannot add anything else but to replicate those considerations. Personally, I cannot give my consent to publish until those comments are seriously considered and the paper changed accordingly.

Author Response

I noticed that no one of my previous annotations (i.e., distributions, limitations, references) have been taken into serious consideration and the paper has not changed in response to them. I cannot add anything else but to replicate those considerations. Personally, I cannot give my consent to publish until those comments are seriously considered and the paper changed accordingly.

Once again, I thank you for your annotations on my manuscript. I wanted to assure you that I took your comments in serious consideration and I have addressed them in the first revision. I understand that there are some aspects that might not have been clear, so I added clarifying statements in the manuscript about the limitations of the study, choice of distributions, and references to justify these choices in this new revised version. You can see the changes that I made from the first and second revisions in the manuscript version with the tracked changes. The line numbers are based on the clean version of the manuscript which can also be found in the PDF.

“For example, one of the main findings of the paper is that, independent of the vaccination status  fo a population, an emergent strain will become seroprevalent very fast. This is very counterintuitive. I suppose there could be be explanations (both at a mathematical and biological level) that should be discussed, and clearly emphasized as LIMITATIONS of the paper”

Thank you for these comments. The results are counterintuitive at first glance, but we need to recognize that one of the assumptions of the model is that the emergent strain is fully resistant to the immunization provided by the vaccine. The vaccine was made to control the infections of the current strain in the population, which was shown to decrease after the emergence of the new strain. An example diagram is included in Figures 2, 3, and 4 in the revised manuscript. The rapid dominance of the emergent strain within the infected should not be perceived as the failure of a vaccine but rather, it is a by-product of the effectiveness of the vaccine. In the context of the model presented, vaccination allows for individuals to only be susceptible to emergent variants that are not covered by the available vaccine, and hence the faster dominance. This point was discussed further in Lines 361-378 in the Discussion Section of the paper. Lines 379-394 relates the results obtained from the simulation-based experiment to the context provided in Section 1 regarding the emergence of highly infectious variants (VUM) and strains that are vaccine-resistant. I emphasized in Lines 406-408 that these results are observed for the specific implementation scheme of vaccination in the population to motivate future study in TTD for different variations of SIRV models.

“1) the simulation presented in this paper, with its underlying stochastic model, has been not adequately designed. For example, there is scarce information about the distributions on which the SIR model is based. those distributions are neither clearly discussed nor motivated. I take as example a paper where changing from a Poisson to Negative binomial did not change the results of the model which was robust enough to provide evidence in support of the hypothesis:AA.VV. Reopening Italy’s schools in September 2020: a Bayesian estimation of the change in the growth rate of new SARS-CoV-2 cases. BMJ Open 2021;11:e051458. doi: 10.1136/bmjopen-2021-051458. I would like the authors discuss on this point, hopefully taking inspiration from the mathematical modeling of the above paper which was robust to distribution change.”

Lines 97-102 addresses the limitation of the stochastic SIR model used in the study by specifiying the discretization scheme of the continuous time Markov chain implementation of the infection, recovery, and vaccination transitions in the model. We explicitly point out that we are going to utilize the scheme presented by Breto et. al., which involves using an Euler implementation scheme to simulate these transitions. The scheme presented by Breto et. al. can be implemented using Poisson distribution, binomial/multinomial distributions.

The choice of distributions for the transition was discussed further in lines 153-167, alongside references that used these implementation schemes for compartmental models. The reason for not using the Poisson distribution is discussed in Line 156-157, where the unbounded nature of the Poisson distribution could lead to negative populations in some simulations.  Lines 168-170 shows that the binomial/multinomial and Poisson schemes agree for small time increments with the appropriate references, so we would be able to observe the same behavior of the TTD for Poisson distributed transitions without dealing with the unbounded nature of the Poisson distribution.

I have also emphasized the limitation of the results to the specific discretization scheme presented in the study in the Discussion section in lines 406-408.

2 I think that the results the authors have achieved can depend on the type of vaccination they have in mind. Maybe, different results could be achieved depending wether the vaccine provides a sterilizing immunity or not (like in the case for COVID). This point should be clearly discussed.

Thank you for this interesting comment. A revision that was carried over from the first revision was the emphasis that I was interested in a macroscopic view of the spread of infections and that studying the intra-host mechanism was outside the field of the study (Lines 97-98). The study only assumes that the vaccine provides complete and unwaning immunity in the population, which was stated in Section 2.1.1.

One factor that could be related to the type of vaccination is the vaccination rate, which is the rate at which vaccinated individuals develop immunity from the current existing strain. The effect of the vaccination rate is higher when there is low vaccine coverage and the emergent strain has relatively low transmissibility. Refer to Lines 352 to 378 in discussing the implications of the vaccination in the population in the simulations using the provided SIR model.

In conclusion, I say that some results are so specific that a detailed discussion that better motivates the possible causes is mandatory before publication. Hence major revision.

Reviewer 3 Report

Well done with the corrections

Author Response

Thank you very much for your valuable input!